# Ultra Short Course Chemotherapy for Early-Stage Non-Hodgkin’s Lymphoma in Children

**DOI:** 10.3390/children9091279

**Published:** 2022-08-25

**Authors:** Elisabetta Schiavello, Filippo Spreafico, Francesco Barretta, Giulia Meraviglia, Veronica Biassoni, Monica Terenziani, Luna Boschetti, Giovanna Gattuso, Stefano Chiaravalli, Luca Bergamaschi, Nadia Puma, Giovanna Sironi, Olga Nigro, Marta Podda, Cristina Meazza, Michela Casanova, Andrea Ferrari, Roberto Luksch, Maura Massimino

**Affiliations:** 1Pediatric Oncology Unit, Fondazione IRCCS Istituto Nazionale dei Tumori, 20133 Milan, Italy; 2Department of Clinical Epidemiology and Trial Organisation, Fondazione IRCCS Istituto Nazionale dei Tumori, 20133 Milan, Italy; 3Department of Pediatrics, Vittore Buzzi Hospital, University of Milan, 20122 Milan, Italy

**Keywords:** non-Hodgkin lymphoma, early-stage, chemotherapy, late-effects, intrathecal, maintenance

## Abstract

Early-stage non-Hodgkin’s lymphomas (ES-NHL) are associated with high survival rates. To minimize the risk of long-term sequelae, the duration and intensity of chemotherapy have been progressively reduced. Between 1988 and 2018, children with ES-NHL were treated at a single institute with two subsequent protocols. Protocol I consisted of a 7-week induction phase followed by a maintenance phase alternating 6-mercaptopurine plus MTX, a brief reinduction, and thioguanine plus cytosine arabinoside, for a total duration of 8 months. The subsequent protocol II (applied since 1997) was modified adding etoposide plus a further dose of HD-MTX and omitting maintenance in all histological subtypes except T-lymphoblastic lymphoma (T-LBL), for a total duration of 9 weeks. Intrathecal prophylaxis was not provided in either protocol. With a median follow-up of 98.4 months, the 5-year event-free survival (EFS) rates in protocol I (*n* = 21) and II (*n* = 25) were 76.2% and 96%, respectively, and the 5-year overall survival (OS) rates were 90.5% and 96%, respectively. None of the patients experienced disease progression or relapse within the central nervous system (CNS). Acute toxicity was manageable in both protocols, except for a case of presumed acute cardiotoxic death; no chronic sequelae were evident. Low-intensity chemotherapy for 9 weeks without intrathecal prophylaxis was sufficient for curing children with ES-NHL, without jeopardizing the excellent survival rate of this disease.

## 1. Introduction

Evolution in the treatment of pediatric non-Hodgkin lymphoma (NHL) is an important success of recent decades: before the 1970s, less than 20% of children with NHL survived; nowadays, over 80% of children with NHL can be cured, including those with disseminated disease at diagnosis [1].

Early-stage NHLs represent one-third of all pediatric NHLs and are characterized by an excellent survival rate, regardless of histological subtype. This has led clinicians to progressively omit radiotherapy and to use less intensive chemotherapy regimens [2,3,4,5,6,7]. The five-year EFS rate for localized non-lymphoblastic NHLs ranges between 88 and 98% in the largest trials of the German Berlin–Frankfurt–Munster (BFM) group [8], the American Children Cancer Group [9], the Pediatric Oncology Group (COG) [6], the French Society of Pediatric Oncology (SFOP) [10], and the international French–American–British (FAB-LMB) [11] cooperation. Survival for T-LBL is lower but still high, especially after adopting longer and more intensive regimens similar to those for advanced stages [12]. Furthermore, localized T-LBL is a rare entity and perhaps represents a failure to detect dissemination rather than a truly localized disease, thus making larger analyses even more difficult.

The focus of pediatric oncologists has now shifted to minimizing acute toxicity and long-term sequelae (i.e., second cancers, cardiomyopathies, endocrine dysfunctions, and infertility), without jeopardizing the excellent survival rates of this disease. We modified treatments offered at our institution from long-lasting ones to short courses of therapy back in 1988 [13].

We herein present the results of an analysis of 46 children with ES-NHL treated at our institute according to two different protocols, with the aim of comparing the efficacy and toxicity between two approaches that differ in treatment intensity and duration.

## 2. Material and Methods

### 2.1. Patients

Children and adolescents up to 20 years of age with ES-NHL of any histological subtype treated at our institute were eligible for this retrospective study. Informed consent was obtained from parents or patients over 18 years old, in accordance with the Declaration of Helsinki. The protocol was approved by the institutional ethical board.

All diagnoses, when performed in other centres, were reviewed centrally. For the purpose of this paper, lymphomas were classified using the 2016 WHO revision [14].

The pretreatment staging workup included, in addition to complete blood analysis and physical examination, chest X-ray imaging, ultrasonography, a computed tomography (CT) scan of the chest, abdomen and pelvis bone marrow aspirates and biopsies, cerebrospinal fluid analysis, and nuclear medicine imaging (gallium scintigraphy or positron emission tomography (PET), depending on the period). Magnetic resonance imaging (MRI) was used to evaluate specific sites of disease, such as the head and neck. Disease staging was performed according to the Murphy–St. Jude staging system [15] and the definition of early-stage NHL (ES-NHL) included stages I and II. Each patient was screened for Epstein–Barr virus (EBV) infection status at diagnosis. A human immunodeficiency virus (HIV) antibody test was not routinely performed but only in the case of a known or suspected family history for HIV infection.

Fifty patients with ES-NHL treated between 1988 and 2018 were eligible for this study. Four of the fifty patients were excluded from the analysis as they did not receive further treatment after a complete surgical resection of their tumors: 3 single cutaneous localizations of anaplastic large cell lymphoma (ALCL) in 2 patients, subcutaneous panniculitis-like T-cell lymphoma in 1, and testicular diffuse large B-cell lymphoma (DLBCL) in 1. None of the three surgically treated patients had relapses. The remaining 46 patients were treated according to two subsequent protocols: 21 between 1988 and 1997 with protocol I, and 25 between 1997 and 2018 with protocol II.

### 2.2. Treatment

Initial surgery was recommended for diagnostic purposes, considering complete tumor excision if feasible without risks.

The protocol scheme is detailed in Figure 1. Briefly, protocol I was divided into 7 weeks of induction phase followed by a maintenance therapy, for a total duration of 8 months. This maintenance phase consisted of alternating oral 6-mercaptopurine plus MTX, a brief reinduction (with cyclophosphamide, vincristine, and doxorubicin), and oral thioguanine plus cytosine arabinoside. Involved-field radiation therapy was considered only for patients with no complete remission after week 4 of induction.

In protocol II, the maintenance phase was abolished in all subtypes except for T-LBL. The induction phase was similar to the previous one: the only difference was a final intensification with vincristine plus high-dose MTX and etoposide (in weeks 8 and 9), for a total duration of only 9 weeks. In the event of no complete remission after week 4 of induction, an intensification was evaluated.

In both protocols, intrathecal chemotherapy was not provided in all patients.

### 2.3. Response Criteria and Follow-Up

Tumor response was assessed by comparing imaging prior to treatment, after week 4, and at the end of induction; RECIST criteria were adopted in the retrospective re-evaluation [16]. Subsequent follow-up was performed at 4- to 8-week intervals during the first year after diagnosis, every 6 months for the next 3 years, and then every year, for a total of 10 years.

### 2.4. Statistics

The primary endpoints of the present study were OS and event-free survival (EFS). OS was defined as the time between diagnosis and death for all causes, with censoring at the last follow-up date for patients who were alive. EFS was defined as the time between diagnosis and first event (disease progression or relapse, or second malignancy) or death for all causes, whichever occurred first, with censoring at the last follow-up date for patients alive and event-free. OS and EFS curves were estimated with the Kaplan–Meier method and statistically compared with the log-rank test to investigate associations between the endpoint and the clinicopathologic characteristics and treatments, including protocol (I, II), sex (male, female), lactate dehydrogenase (LDH) level (high, normal), histology (LBL, other), stage (Murphy I-II), and surgery (biopsy only, complete resection). Two univariable Cox models were performed to assess the association between age (modelled using 3-knots restricted cubic splines) [17] and the endpoints. The median follow-up was estimated with the reverse Kaplan–Meier method using OS data [18]. Differences in the median (numerical variables) and proportions (categorical variables) between the characteristics of patients treated with protocol I vs. II were compared using standardized mean difference (SMD) [19].

The analyses were performed with SAS^TM^ (Cary, NC, USA) and R software (R Foundation for Statistical Computing, Vienna, Austria).

## 3. Results

### 3.1. Patients

Of the 46 evaluable patients, 25 were male and 21 female; the median age was 10.3 years (range: 2.2–19.8) (Table 1). One patient was HIV-positive. The primary disease site was the Waldeyer ring in 13 patients (28%), lymph nodes in 11 (24%), the ileocecal region in 10 (22%), and other sites in 12 (26%; stomach [*n* = 4], bone [*n* = 3], skin [*n* = 2], ovary [*n* = 1], anal region [*n* = 1], conjunctiva [*n* = 1]). The diagnoses were: Burkitt lymphoma (BL) in 25 cases (54%), DLBCL in 8 (17%), precursor B-cell lymphoblastic lymphoma (B-LBL) in 5 (11%), lymphoma originating from mucosa-associated lymphoid tissue (MALT) in 4 (9%), ALCL in 2, not-otherwise-specified high-grade B-cell lymphoma in 1, and pediatric-type follicular lymphoma in 1. No patients with early-stage T-LBL were enrolled in both protocols. Twenty-seven patients had stage I (59%) and 19 had stage II (41%) disease. Ten patients had elevated LDH levels (range: 507–1048 IU/L; cut-off: 500 IU/L) whereas B-symptoms were reported in three patients only. At the beginning of chemotherapy, 28 patients (61%) did not have macroscopic evidence of disease, due to a complete surgical excision; the remaining 18 patients received only a diagnostic biopsy or partial surgery. No clinically relevant differences were observed between the two groups of treatment.

### 3.2. Response to Treatment

At the radiological evaluation at week four, CR was achieved in 18/21 patients in protocol I and in 23/25 patients in protocol II. One patient, in protocol II, could not be evaluated for CR due to early death (24 h after the second cycle of doxorubicin), probably due to an acute cardiotoxic event. Of the four patients who did not reach a complete response to treatment, one (in protocol I) experienced an early local disease progression after the first week of therapy and was successfully treated with the institutional protocol for advanced-stage disease [20]. The remaining three patients (two in protocol I; one in protocol II) had radiological evidence of residual disease at the end of induction: one was treated with local radiotherapy (25.2 Gy) and the other two with chemotherapy intensification according to the protocol for advanced-stage disease [20]; all three patients achieved CR using these strategies.

### 3.3. Disease Relapse

Two patients had local relapse; both occurred one month after the end of induction in protocol I, and they had reached CR at the intermediate re-evaluation. One patient with relapsed ileocecal disease received institutional treatment for advanced-stage disease but, after a transient remission, died three months after relapse; the other patient, with cervical nodal disease, was successfully treated with the protocol for advanced-stage disease [20] plus local cervical radiotherapy (35 Gy).

### 3.4. Survival

After a median (first and second quartile) follow-up of 98.4 months (65.1–121.8), 4/46 (8.7%) patients had died. One patient died due to relapsing BL; the other three patients were all affected by B-LBL: one died probably due to cardiac arrest after doxorubicin was administered (week five, protocol I), one developed a paraspinal Ewing’s sarcoma ten years after the diagnosis of lymphoma, and the fourth patient, with a previous HIV infection, died from lung tuberculosis three years after the end of lymphoma therapy. These three patients with B-LBL were previously successfully treated with our protocols (two with protocol I and one with protocol II). The 5-year OS (95% CI) and EFS for the whole cohort were 93.3% (86.2–100%) and 86.6% (77.2–97.2%), respectively; the 5-year EFS and OS rates divided into protocols I and II were 76.2 and 96% (EFS) and 90.5 and 96% (OS), respectively (Figure 2).

At univariable analysis, only B-lymphoblastic histology vs. other histology showed a significant negative impact on OS and EFS, but it has already been pointed out that three out of four patients who died had B-LBL histology and the cause of the death was unrelated to lymphoma. Five-year OS and EFS were 60% (29.3–100%) vs. 97.6 (93.0–100%; *p* < 0.001) and 60% (29.3–100%) vs. 90.1 (81.4–99.8%; *p* = 0.005), respectively (Figure 3). Protocol II (vs. protocol I) and female (vs. male) sex showed a protective effect only for EFS, with 5-year estimates of 96.0% (88.6–100%) vs. 76.2% (60.0–96.8%; *p* = 0.026) and 100% vs. 75.6% (60.3–94.7%), respectively (Figure 4). None of the patients treated with protocol II experienced disease progression or relapse and no cases of CNS disease progression or relapse were observed at all. It is worth mentioning that patients with both resected and unresected mass received the same treatment and there was no significant difference in the outcome between the two groups (5-yr OS and EFS 92.4% vs. 94.4% and 85.1% vs. 88.9%, respectively; Figure 5).

### 3.5. Toxicities

Considering the induction phases, there were 39 episodes of grade 3–4 toxicity according to the Common Terminology Criteria for Adverse Events: 20 episodes of hematologic toxicity (51%), 18 episodes of liver toxicity (46%) associated with MTX administration, and one infection. Moreover, 19 patients (41.3%) presented at least one episode of grade 1–2 peripheral neurotoxicity associated with vincristine administration that in 11 cases required modification of the subsequent doses of the drug. In the maintenance phase, there were only seven episodes of grade 3–4 toxicity: six episodes of hematologic toxicity and one infection. No patients required dialytic treatment. All haematological toxicities were reversible. Overall, both the protocols were well tolerated except for the one patient already described who died due to cardiac arrest 24 h after the administration of doxorubicin (25 mg/sqm, with a 3-h intravenous infusion; 50 mg/sqm cumulative dose up to that point of cyclophosphamide and vincristine administration (week 5)); an autopsy was not performed, but no cardiac disease was evident at the baseline echocardiographic evaluation. With a median follow-up of 98.4 months, we did not detect any late effect associated with treatment; no cases of late doxorubicin-induced cardiotoxicity have been reported to date. The patient that developed a paraspinal Ewing’s sarcoma ten years after the diagnosis of precursor B-cell lymphoblastic lymphoma had no evidence of a predisposing familiar cancer syndrome, and this event does not seem correlated with the previous chemotherapy.

## 4. Discussion

Non-Hodgkin’s lymphomas account for 7% of all childhood tumors [21] and localized NHLs (Murphy stage I and II) represent only one-third of all new diagnoses of NHLs and are associated with excellent survival rates of above 90% in most recent studies [1]. The treatment of early-stage NHL has evolved over recent decades, first omitting radiotherapy and later attempting to decrease the intensity and duration of chemotherapy, in order to reduce acute toxicities, hospitalizations, and long-term sequelae. In this study, we observed that long-term OS results of a short course of chemotherapy were comparable with more intensified therapies that we had used before in 1988 [13].

These results compare well with published reports; in particular, between 1983 and 1991 the American Pediatric Oncology Group conducted two consecutive trials for children with any subtype ES-NHL, demonstrating that radiotherapy could be safely avoided in all patients and that, in the case of non-lymphoblastic lymphoma, there was no difference in the OS with patients receiving 9 weeks versus 8 months of therapy [5,6]. However, patients with lymphoblastic histology had a poorer outcome (EFS 63%) after shortened treatment, even though the overall outcome was the same thanks to the salvage therapy [6].

Recently, most of the European and International groups have been trying to stratify patients in risk groups based on stage, resection, and tumor burden (as LDH level), and to treat them depending on the risk group. In the SFOP LMB89 trial, patients with resected stage I and II (group A) received two courses of COPAD (cyclophosphamide, vincristine, prednisone, and doxorubicin), while patients with non-resected stage I and II (group B), after a cytoreductive prephase, received two courses of COPADM (similar to the previous course with the addition of HD-MTX), two consolidation courses with cytarabine and HD-MTX, and a maintenance with cyclophosphamide, vincristine, doxorubicin, HD-MTX, and prednisone. Intrathecal chemotherapy with MTX and cytarabine was provided only for group B. With this strategy, the 5-year EFS was 98% for group A and 92% for group B [10]. Subsequently, the international FAB/LMB96 study aimed to determine whether the excellent result reported by SFOP could be reproduced in a larger group of patients. They demonstrated that early-responding patients in group B could further reduce therapy [11,22]. In the BFM95 study, patients with resected stage I and II (group R1) received two courses of chemotherapy, including (in course A) dexamethasone, ifosfamide, methotrexate, cytarabine, and etoposide, and (in course B) dexamethasone, cyclophosphamide, methotrexate, and doxorubicin. Patients with non-resected stage I and II (group R2), after a cytoreductive prephase, received courses A and B twice, with the addition of vincristine. Moreover, both of the two courses included a triple-drug intrathecal chemotherapy. With this treatment, the 5-year EFS was 100% for the group R1 and 96% for the group R2 [8,23]. The AIEOP (Associazione Italiana Ematologia Oncologia Pediatrica) group achieved a 5-year 100% result for group R1 and 86.9% for group R2 with a modified version of the BFM protocol [24]. In our cohort, we did not observe any localized T-LBL, but we had five patients with early-stage B-LBL that were all successfully treated with our protocols; unfortunately, three of them died for causes unrelated to lymphoma. In Protocol II, we did not plan any maintenance treatment for ES B-LBL and we reserved it only for T-LBL but none were enrolled. Therefore, it is very difficult to comment on the real value of a maintenance treatment in ES B-LBL, as it is a rare entity, and our series is too narrow to draw any conclusion regarding the best treatment for this specific histological subtype. Considering patients with localized lymphoblastic lymphoma, prolonged regimens based on those used for acute lymphoblastic leukemia have shown to be more effective. In particular, the BFM90 trial provided a 9-week induction course with eight drugs, followed by a consolidation phase with HD-MTX and maintenance, for a total duration of 24 months [25]. In NHL BFM90 [8], NHL BFM95 [23], and EURO-LB 02 [12], patients were stratified according to stage. Patients with limited stage I/II disease did not receive reintensification treatment, and the outcome analysis of trial EURO-LB 02 supported using the stage of disease as a stratification criterion for B-LBL resulting in a favorable EFS for B-LBL with limited disease representing almost half of B-LBL patients. Furthermore, in T-LBL it is well known that the number of patients with limited I/II disease is very low; therefore, the stage of disease is an insufficient parameter to identify low-risk T-LBL patients potentially available for treatment de-escalation. On the basis of these large prospective trials, it is now clear that a maintenance is useful in all lymphoblastic lymphomas, but an in-depth study on the role of de-escalation in localized B-LBL has not yet been carried out and future trials could shed some light on this.

In cancers with a high cure rate, such as ES-NHL, it is crucial to consider the potential of severe late effects. In our protocols, intrathecals were safely omitted without registering any CNS relapse, since high-dose MTX was considered sufficient for CNS prophylaxis. Furthermore, maintenance chemotherapy was safely removed for the non-lymphoblastic type. With protocol II, we achieved a high survival probability (5-year EFS and OS 96%), comparable to that obtained in international trials. No cases of progression or relapse were reported. Only one patient had radiological persistence of the disease at the end of treatment, and they were successfully cured with the more intense protocol used for advanced stages.

The small cumulative doses of protocol II, with cyclophosphamide (2.5 g/sqm), doxorubicin (75 mg/sqm), etoposide (500 mg/sqm), and methotrexate (350 mg/kg) are quite far from the universally recognized toxicity thresholds [26,27] that reduce the risk of acute toxicities and long-term sequelae as well as the number of hospitalizations and the effective cost of the treatment. Apart from HD-MTX, all the other courses could be administered in the outpatient clinic. The regimen was well tolerated and, considering the overall population, most patients did not present late effects associated with therapy.

It is worth mentioning that, unlike in other studies, patients with both resected and unresected mass received the same treatment, and there was no significant difference in the outcomes between the two groups, as shown in Figure 5. This result confirms that the quality of surgery plays a marginal role in the treatment of ES-NHL; we recommend it being performed only for diagnostic purposes and if complete resection can be accomplished with no risk. In addition, intrathecal prophylaxis was safely omitted in all patients, even in those with higher risk localizations (such as the head and neck).

The limitation of our study was the small size of the population, the retrospective design, and the lack of a systematic assessment of fertility that could be a major concern.

In conclusion, nine weeks of chemotherapy with low cumulative drug doses are sufficient to cure childhood ES-NHL. A significant improvement in our understanding of the molecular pathogenesis of malignant transformation will enable us to further optimize the therapeutic index of current therapies.

## Figures and Tables

**Figure 1 children-09-01279-f001:**
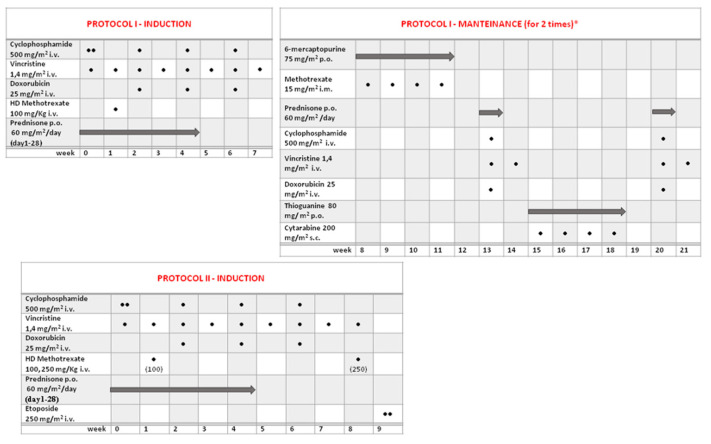
Protocol I scheme with induction (7 weeks) and maintenance (8–21 weeks for 2 times); protocol II scheme with induction (9 weeks) and the same maintenance as protocol I only for T-LBL histology.

**Figure 2 children-09-01279-f002:**
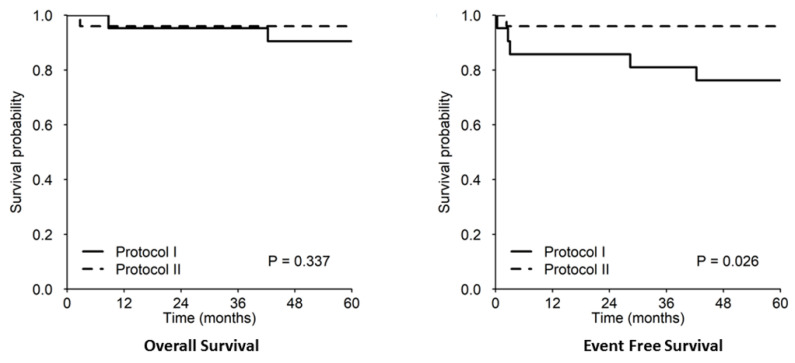
The 5-year OS/EFS rates in protocols I and II.

**Figure 3 children-09-01279-f003:**
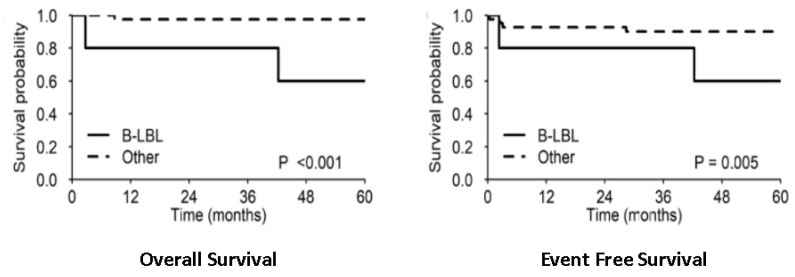
The 5-year OS and EFS rates in B-LBL histology vs. other histology.

**Figure 4 children-09-01279-f004:**
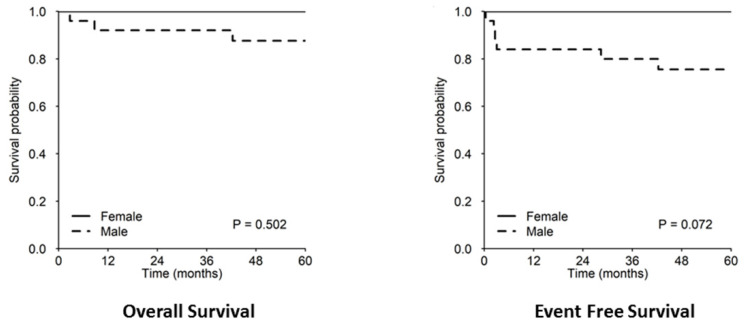
The 5-year OS/EFS rates in females vs. males.

**Figure 5 children-09-01279-f005:**
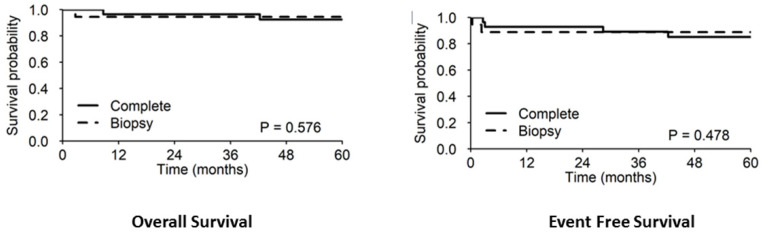
The 5-year OS/EFS rates in completely resected and only biopsied patients.

**Table 1 children-09-01279-t001:** Characteristics of patients.

	Protocol I	Protocol II	All
No. of patients	21	25	46
Median age, years (range)	10, 9 (2.2–18.5)	9, 2 (4.3–19.8)	10, 3 (2.2–19.8)
Male:Female	12:9	13:12	25:21
“Resected” tumor at diagnosis	12	14	26
**Histological Classification**			
Burkitt L	10	15	25
Precursor B lymphoblastic L	3	2	5
MALT L	3	1	4
DLBCL	3	5	8
ALCL	2	0	2
Follicular lymphoma	0	1	1
Not-otherwised-specified high–grade B lymphoma	0	1	1
Relapses/failure to induction	5	2	7
Death related to L	1	0	1
Death for other causes	2	1	3

Abbreviations: L, lymphoma; MALT, mucosa-associated lymphoid tissue; DLBCL, diffuse large B-cell lymphoma; ALCL, anaplastic large cell lymphoma.

## Data Availability

The data presented in this study are available on request from the corresponding author. The data are not publicly available due to privacy statements made in informed consent obtained from both the participating healthcare personnel and parents of the studied children.

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
