# Peer review of "Ultra Short Course Chemotherapy for Early-Stage Non-Hodgkin’s Lymphoma in Children"

_children, 2022, doi:10.3390/children9091279_

Round 1

Reviewer 1 Report

Dear authors,

Thank you for a very interesting, relevant, and well-written article. Before publication, I would like to see or have a discussion about whether each patient was screened for HIV and EBV. I have no further comments.

Best of luck.

Author Response

Thank you for your comment; each patient was screened for Epstein-Barr virus infection status at diagnosis. HIV antibody test was not routinally performed, but only in case of family history known or suspected for HIV infection. We don’t perform HIV test routinely even today but only in the case of special needs to cryopreserve material or stem cell harvest. For the patient with HIV, the positivity was already known at the time of lymphoma  diagnosis.

We added this comment from line 95.

Reviewer 2 Report

The manuscript of original research by Elisabetta, et al. titled “ULTRA SHORT COURSE CHEMOTHERAPY FOR EARLY-STAGE NON-HODGKIN’S LYMPHOMA IN CHILDREN” describes outcome of children with early-stage (ES) non-Hodgkin lymphoma using two different treatments according to the treatment era in a single institution. First of all, the title is written in large letters only but it should observe the regulation of Children. Authors’ affiliation number should be superscript.

Because of retrospective nature of this study, there are several limitations as authors discuss in the article. The biggest concern is the small number of the patients. That is especially why I must insist that authors should pay attention to how to describe the data in Results, as well as how they interpret the data in Discussion. The authors showed a total of 4 pairs of Kaplan-Meyer curves, which they must believe very important. I do not understand why Figure 5 is located and explained in Discussion only. These figures should be explained in detail in Results and also logically discussed in Discussion.

T-LBL seems to be excluded from the study population. But this fact is not explained well. In abstract, there are sentences “The subsequent protocol II (applied since 1997) was modified adding etoposide plus a further dose of HD-MTX and omitting maintenance in all histological subtypes unless T-lymphoblastic (T-LBL), for a total duration of 9 weeks.”. Readers cannot understand how authors handled T-LBL (no experience of localized T-LBL, experienced but not treated per protocol I nor II, or treated by the protocol but excluded out of the analyses). If they experienced localized T-LBL during the study period, the number should be included in Table 1 and they should explain how they handled the population in the analyses.

On the other hand, B-LBL are included. Because some reports support the application of maintenance chemotherapy for B-LBL, discussion on this point is very important. However, authors did not try to discuss in lines 348-351. I believe that discussion regarding +/- maintenance for B-LBL is inevitable even if they have a few patients.

I cannot find the definition of early-stage (ES) but can find the definition of “localized” in line 93. Then, in the next line, they use the term “ES-NHL”. Please specify the definition because they are confusing.

Last but not least, please consider to make minor amendments.

Line 24. Double spaces before “thioguanine”?

Line 216. Double spaces before “divided”?

Line 255. “98,4 months” should be “98.4 months”?

Line 274. “american” should be “American”?

Line 322. Comma after “Futhermore”?

Author Response

Thank you for your constructive comments in particular regarding the clarification on lymphoblastic lymphoma, in fact it was necessary to point out better that in the range time of our two protocols we didn’t diagnose any early stage T-LBL, although the protocol criteria included this histotype; however, it seemed correct to add that in the initial project the Protocol II was designed with the addition of the maintainance only for  early stage T-LBL; we have probably not diagnosed early stage T-LBL due to the rarity of its limited-stage presentation. 

Point by point: 

  • We have corrected the title with lowercase letters and Authors’ affiliation number as suggested.
  • We have added in the Results a comment about Figure5 concerning the Kaplan-Meyer curve in which there is no difference in survival for completely resected and only biopsied patients. We have added this part from line 149 (in Results-Survival). We have omitted survival percentages at line 410, already written in the Results.We have therefore left in the Discussion the already written comment that quality of surgery plays a marginal role in the treatment of ES-NHL and the recommendation to perform surgery only for diagnostic purposes, and if complete resection can be accomplished with no risk (line 402-408).
  • As already written above we didn’t diagnosed any limited stage T-LBL probably due to its rarity, although the study design had foreseen that it could be included. We have corrected this concept in the manuscript in several points and focused the discussion only on five patients with B-LBL: all were successfully treated with our protocols; unfortunately three of them died for causes unrelated to lymphoma.
  • We have added in the Discussion (from line 347) a comment on treatment reduction in International Trials (BFM90, BFM95 and EURO LB 02) for patients with early stage B-LBL and the decision for our study to omit maintenance; our five patients with B-LBL were successfully treated with our treatment but we know that our series is too small to draw any conclusion regarding the best treatment for this specific histological subtype.

See added comments on the previous 2 points at lines: 142, 185, 226, 231-233, 238-241, 352-360, 366-384, (lines 417-420 were deleted).

  • The definition of early stage NHL includes all stage I-II NHL according to Murphy- St Jude staging system; thanks for the comment: the  “localized” definition is confusing. (line 94)
  • We have added minor spelling/grammar amendments in lines: 25, 103, 141, 314.

Round 2

Reviewer 2 Report

The second version of the manuscript has been improved with additional discussion and corrections of some typos. However, several typos still remain. After collection of all of the following and potentially existing others, I would like to accept the manuscript. However, the correction should be stricty examined and confirmed by the editorial team.

Line 90: double space before “early-stage”

Line 117: double comma after “T-LBL”

Line 223: double space after “developed”

Line 228: add space before “The 5-year”

Line 345: check the indent before “five patients”

Line 347: add comma after “In Protocol II”

Line 369: double space after “B-LBL”

Line 395: the sentence is incomplete. How about adding “as shown in” before “Figure 5”?

Table 1: In “Median age, years (range)”, I recommend to use “.” instead of “,”.

Author Response

Thank you for your comments;

I corrected all reported typos:

  • Line 90; 117; 223; 228; 345; 347; 369; 395; Table 1
  • I noticed another typo: line 348: I added a space between but and none

Thank you,

best regards,

Elisabetta Schiavello